# Visitor Perceptions and Effectiveness of Place Branding Strategies in Thematic Parks in Bandung City Using Text Mining Based on Google Maps User Reviews

**Munawir [1,2,\*]** **, Mochamad Donny Koerniawan [3]** **and Bart Julien Dewancker [4]**

1    Faculty of Environmental Engineering, The University of Kitakyushu, Kitakyushu 808-0135, Japan
2    Faculty of Computer Engineering, The University of Cokroaminoto Palopo, Palopo 91921, Indonesia
3    Department of Architecture, Bandung Institute of Technology, Bandung 40132, Indonesia;
     donnykoer@gmail.com
4    Department of Architecture, The University of Kitakyushu, Kitakyushu 808-0135, Japan;
     bart@kitakyu-u.ac.jp
\*    Correspondence: munawiradam2@gmail.com; Tel.: +81-902-8540-585

**Abstract:** The city of Bandung, Indonesia contains thematic parks which use certain themes to highlight the features of the park. They are also used as a branding strategy for the city as a whole. As social networking has become a type of media used by most global populations to share experiences and stories and to influence perceptions, and because online reviews are one way to get potential positive information about the success of a business or service, we analyzed online reviews from the Bandung thematic parks. We identified that thematic parks have an influence on the branding of the city of Bandung. Data collection involved data extraction from Google Maps user reviews. Text mining was used to collect the information attributes needed to determine the public perceptions of thematic parks. Data analysis was used to determine the extent to which a park can be a benchmark for place branding in Bandung. This research found that the influence of the thematic park concept is a good strategy for the city of Bandung. Online reviews show that thematic parks in Bandung are better known than non-thematic parks, and that thematic parks get very good ratings and good opinions from online reviewers. This information is expected to be a reference for developing the concept of thematic parks, especially in the city of Bandung, and it can be used by the government, architects, and urban designers to get a better understanding of the users' perceptions and as a benchmark for similar projects.

**Keywords:** text mining; online reviews; urban park; urban landscape; city branding; place branding; thematic park

## 1. Introduction

The benefits and functions of green space for humans can be direct or indirect. In many studies of green spaces, parks and other green areas have been shown to provide benefits by making a city more livable and sustainable [1]. Urban parks provide a multiplicity of benefits to their communities. They create recreation opportunities, preserve open spaces and wildlife habitats, beautify neighborhoods and sections of cities, serve monumental or memorial functions, provide visual diversity, act as landmarks, and even guide traffic flow [2]. Parks are a destination where members of the community choose to spend their free time, because no money is required to enjoy them, and because they are comfortable, open spaces [3]. The existence of a city park [4] is an important part

of an ecosystem's complex urban network that provides important services. The uses of city parks include environmental, aesthetic, recreational, psychological, social, and economic aspects [5].

Nowadays, cities, regions, and even countries across the world are developing strategies to develop their competitive advantage over others [6]. They use branding strategies that have grown in the last decade. These branding positions are intended to promote their uniqueness among growing competition for capital, visitors, residents, and corporations. Nevertheless, branding positions are not only used by global cities, capitals, and tourist destinations, but also by smaller growing cities and even urban parks.

One of the concepts used to make parks known to people is place branding, a method initiated by the city of Bandung. The provision of parks in the residential areas of the city of Bandung has experienced a radical paradigm shift: parks have become a key attraction at a city service scale and provide sources of entertainment and recreation for urban communities through their new physical designs and attractive facilities [7–9]. Revitalizing public spaces into several thematic parks has helped Bandung stand out from other cities and has improved the city's branding. Thematic parks are parks with a variety of interesting themes and an artistic atmosphere that is prepared as a creative space [10]. Since human creative activities are varied, creative spaces can be used to link many different activities with urban spaces [11,12].

Branding is an effort to build the image of products or services according to expectations [13]. The image of a brand is obtained when the audience has a good understanding of the object being represented. Therefore, branding is done by providing adequate information and experience to the public about the object of branding [14]. The term "place branding" has been mixed and matched indiscriminately with other terms, such as place marketing, urban marketing, and place promotion [15]. The marketing of urban places has been practiced since at least the nineteenth century [16], and cities tend to rely on the methods used in the last three decades, when competition for inward investment, tourism revenue, and residents intensified at various spatial scales. The scope and effectiveness of city marketing is largely determined by the selection and application of the appropriate combination of these measures, although by adopting the marketing mix, as suggested by general marketing, we can distinguish between four distinct strategies for place improvement that are the foundations for building a competitive advantage: design (e.g., character), infrastructure (e.g., fixed environment), basic services (e.g., service provider), and attractions (e.g., entertainment and recreation) [17].

Based on previous studies [18], place attachment has two basic forms: as an emotional bond and as a dependence–identity relationship of a place. Place attachment as an emotional bond refers to a relationship that an individual develops with a particular place through repeated positive interactions [19]. Place attachment arises when a setting—such as a local park—is filled with meanings that create or enhance visitors' emotional ties to it [20]. Place attachment involves dependence on the place (place dependence) and its identity (place identity) [21]. The thematic branding concept is assumed to affect visitors and to help the wider community to increasingly recognize the existence of the park. The thematic branding concept used in the parks indirectly creates segmentation for both park users as well as for the activities in the park. The challenge now lies in determining the effectiveness of place branding strategies related to the thematic park concept in Bandung.

Assessments and perceptions of thematic parks are needed to determine the extent to which thematic parks are known to the wider community and the extent to which they have become a type of place branding for the city of Bandung. A large part of the global population is now connected via online social networks on social media, where they share experiences and stories, and consequently, influence each other's perceptions [22]. One way to get information about perceptions and assessments of a place through social networks is through online reviews [23]. We can identify whether branding is successful or not by looking at the users' perceptions, and we can determine the extent to which the brand is known (place ratings) by using the assessment of the users as a benchmark [24]. Social networks can be used for all stages of brand perception with much lower costs compared with traditional marketing and more effective branding strategies [25].

In this research, we used online reviews to determine the perceptions and assessments of thematic parks in Bandung city by visitors. We used user reviews on Google Maps to collect data on visitors' opinions. Opinion mining or text mining was used to analyze and summarize online review texts [26,27]. Text mining refers to the extraction of information from unstructured data, and it is used in many patent research fields, because it can handle with a large amount of text [28]. The aims of this study were to identify the effectiveness of thematic parks in developing place branding in the city of Bandung and to determine the perceptions of the community about thematic parks through social networks. The perceptions were determined by assessing visitors' online reviews provided by Google Maps. This communication mode is considered to be an effective way to spread information widely and publicly. Besides, this kind of review enables visitors to generate public opinion more freely without any restriction that could lead to psychological bias. Furthermore, the utilization of the star symbol that commonly appears in Google online reviews also allows people to easily provide their perception on the visited park. These user-friendly reviews may influence the number of future visitors. The results could be used by the government, architects, and urban designers to allow the design of better parks based on an understanding of the users' perceptions. This information could also be used as a benchmark for similar projects.

## 2. Literature Review

### 2.1. Thematic Parks

Thematic parks [29] are parks created with certain a theme/concept that is used to give the park a unique characteristic. They include certain characters, which allow visitors to interpret a more specific function of the park. The basic characteristics of thematic parks include their function, location, and potential. The added physical attractiveness of thematic parks invites citizens to come and enjoy activities in these public spaces [30]. The thematic concept has been adopted in Bandung, where urban parks have been renovated with thematic designs as a way of revitalizing them. The development of these parks is intended to promote increased interactions with public spaces and to increase the quantity of open green spaces in the city [31]. The theme of a place is developed using unique and distinctive elements; the theme needs to be specific [32]. Thematic parks aim to differentiate themselves from other parks [33]. Successful development of a theme park should further affect visitors' experiences and increase their likelihood of revisiting. According to Ref. [34], theme park operators should aim to attract visitors by providing an increased number of rides that cater to various demographics, ranging from adventurous rides to those for kids. Ref. [35] mentioned that the selection of the theme is extremely important for the operations of the park. In general, theme parks attempt to create an atmosphere that is linked with another place and time, and they usually emphasize one dominant theme around which architecture, landscape, rides, shows, food services, costumed personnel, and retailing are orchestrated. Attachment to the design and space is closely related to the physical setting of the place. According to Ref. [36], the type of physical setting that gives meaning to an individual may vary. It can include aspects of the built environment, such as houses, roads, and special buildings, or the natural environment, such as lakes, parks, forests, and mountains. As mentioned by Ref. [37], the landscape is an important factor in creating the characteristics of a thematic park, and its visual and spatial features have the potential to affect public interest, leading to the creation of attachment relationships by visitors to the physical environment.

### 2.2. The Place Branding of Public Parks

Thematic parks have a similar function to other city parks; the theme of each park is the only concept that sets it apart. These themes represent a type of place branding that was created by the government for open spaces in Bandung. Place branding can be understood as the development of an identity that shapes a place, both geographically and culturally. According to Ref. [38], the branding of a place can add to its appeal and make the public more aware of its location.

Brand image is the manifestation of the overall brand perception [39]. Destinations are treated as the brands of tourism, and destination image perceptions are often analyzed from the demand side viewpoint of tourists visiting the places [40]. Place branding is commonly understood as a general phenomenon involving marketing, branding, promotion, and regeneration of a particular city, region, or location [41]. Place branding can be defined as the planning and execution of the entire process of creating, maintaining, and developing the perception of a city by its potential customers and other stakeholders. Its aim is to influence the attitudes of customers, and it can benefit the development and growth of the city and focus on the value of the city as a whole [42].

A park often becomes one of the signature attractions of a city and it can be used as a prime marketing tool to attract tourists, conventions, and businesses. Regional parks help to shape a city's identity and give residents pride in their city [43]. According to several studies, public parks can become a type of place branding. Ref. [44] carried out a three-facet evaluation of the brand potential of public parks in Hong Kong as a case study. This proposed triangular approach refers to the measurement of three key dimensions of the brand value of a particular resource (i.e., public parks in this case). The measurements indicate a positive attitude towards the brand among stakeholders, such as visitors and local residents. There is a high level of brand value and the resultant brand potential is also great. Ref. [45] shows how national parks have based their strategy on sustainability, and at the same time, they are contributing to sustainable development through environmental protection and biodiversity on ecologically sensitive sites. With good marketing and promotional strategies, the value of the brand destinations in Montenegro can be increased.

*2.3. Visitors' Perceptions*

The importance of places is what connects city branding to cultural geography. Characteristics of identity, differentiation, and personality can be managed to maximize equity and awareness. There is also a focus upon the ever-necessary consumer orientation. From the viewpoint of the end user, a place is seen in terms of the way one senses, understands, uses, and connects to the place [46]. These factors surround the concept and understanding of what perception means. In selecting a destination or park, visitors consider factors like the park's environment, facilities, rides, and location. Perception [47] is the way in which an individual gathers, processes, and interprets information from the environment. Ref. [48] stated that perceptions are the beliefs about what a consumer receives from goods or services.

## 3. Materials and Methods

*3.1. Study Site*

Bandung, the capital city of West Java, is a province of Indonesia and the country's third largest city. According to the 2015 census, it has a population of 2.5 million and an area of 167.45 km$^2$. It has the vision to be a service city that is clean, prosperous, obedient, and friendly [49]. According to the 2011–2031 Regional Spatial Plan, the city of Bandung aims to be a green city in the future, where multiple park elements and green spaces are available. Efforts to meet the green open space goals require the building of neighboring parks, community housing parks, urban parks, sub-district parks, city parks, urban forests, green lanes, cemeteries, river banking, and railway lines. Thematic parks are needed to achieve the green city concept in Bandung. City branding involves the identification of brand attributes of a city in order for it to gain positive perceptions from many audiences [50]. One of the strategies used by the city government of Bandung is to revitalize the city parks by giving them themes, including names and icons that characterize each park. Thematic parks promote a theme or concept using certain characters, so that when people visit the park, they can capture a more specific impression of the park as well as appreciating its beauty [51]. Based on data from the Information Management and Documentation Officer in Bandung, there are 631 parks in the city of Bandung [52]. Figure 1 shows a map of the study sites.

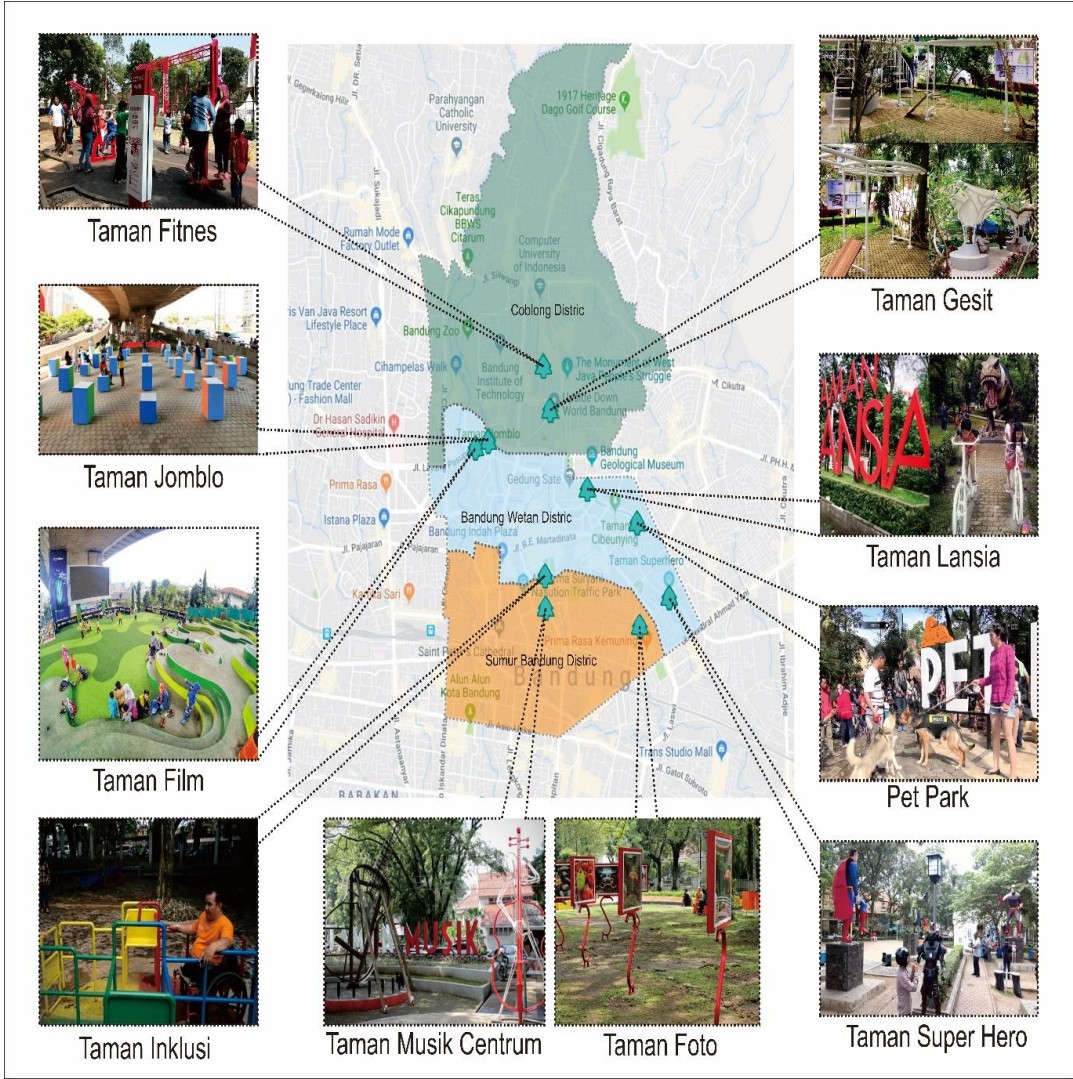

**Figure 1.** Locations of the study sites.

In this study, we considered 10 thematic parks and 15 non-thematic parks in 3 sub-districts in Bandung: Saumur Bandung District, Coblong district, and Bandung Wetan district. Detailed data on the thematic parks and non-thematic parks based in the study locations are shown in Tables 1 and 2.

**Table 1.** Table of thematic parks in Bandung city based on the study locations.

| No | Name of Thematic Park | Area (m$^2$) | District | Descriptions |
|----|----------------------|-------------|----------|--------------|
| 1 | Superhero Park | 2051 | Sumur Bandung | There are several statues of famous superheroes. This park uses superhero statues as thematic icons. |
| 2 | Musik Centrum Park | 2100 | Sumur Bandung | The goal of Musik Centrum Park is to provide a place for residents, especially the youth, to practice or perform music, art, and sport. |
| 3 | Foto Park | 3610 | Sumur Bandung | Foto Park is intended to accommodate photography lovers. In this park, there are several works of photography. |
| 4 | Gesit Park | 755 | Coblong | Gesit Park or Agile park is designed with green and active concepts. The green concept is displayed by a green garden area, while the active concept includes a play area including various sports games. |

**Table 1.** *Cont.*

| No | Name of Thematic Park | Area (m$^2$) | District | Descriptions |
|----|----------------------|--------------|----------|--------------|
| 5 | Fitness Park | 4073 | Coblong | Fitness Park is one of the thematic parks built by the Bandung city government to revitalize the area and provide sports facilities to the public. In accordance with the theme, Fitness Park was specifically designed for outdoor exercise. |
| 6 | Jomblo Park | 1539 | Coblong | Taman Pasupati, better known as Taman Jomblo, is located under the Pasupati bridge. A single person or "jomblo" (in Indonesian terms) is someone who is not in a relationship or is "unmarried". The term "Taman Jomblo" is represented by the presence of a seat in that park that is shaped like a colorful cube with a small size that only fits one person. |
| 7 | Film Park | 1100 | Bandung Wetan | This park is a place of appreciation for Indonesian films. Residents can watch movies from the $4 \times 8$ m Videotron screen with an electrical power of up to 33,000 watts. In accordance with the theme, this park was specifically designed for people to watch films produced by filmmakers from Bandung and also the community. |
| 8 | Lansia Park | 16,257 | Bandung Wetan | Lansia is an abbreviation of Lanjut Usia or "elderly". Lansia Park is a park for the elderly who want to refresh themselves or exercise. Despite its name, the park is visited by individuals of all ages from Bandung or from outside the city of Bandung. |
| 9 | Pet park | 6085 | Bandung Wetan | Animal Park provides a playground for animal lovers and their pets. This park was prepared for the community and animal lovers. |
| 10 | Inklusi park | 2111 | Bandung Wetan | Inklusi Park was developed for disabled people. Inklusi Park is a public facility that was built as part of the effort to reduce discrimination in the city of Bandung. This park is designed to provide a space for disabled individuals to move around and socialize, and it has become a place of healing therapy. |

**Table 2.** Table of non-thematic parks in Bandung city based on the case study.

| No | Name of Park | Area (m$^2$) | Location | District |
|----|--------------|--------------|----------|----------|
| 1 | Maluku Park | 24,023 | Jl. Ambon | Sumur Bandung |
| 2 | Kosambi Park | 759 | Jl. Kosambi | Sumur Bandung |
| 3 | Riau Park | 685 | Jl. Riau | Sumur Bandung |
| 4 | Buton Park | 612 | Jl. Buton | Sumur Bandung |
| 5 | Nias Park | 310 | Jl. Nias | Sumur Bandung |
| 6 | Ganesha Park | 9612 | Jl. Ganesha | Coblong |
| 7 | Panatayuda Park | 2387 | Jl. Panatayuda | Coblong |
| 8 | Bagusrangin Park | 1560 | Jl. Bagusrangin | Coblong |
| 9 | Gelap Nyawan Park | 1656 | Jl. Gelap Nyawan | Coblong |
| 10 | Dayang Sumbi Park | 754 | Jl. Dayang Sumbi | Coblong |
| 11 | Gasibu Park | 25,962 | Jl. Gasibu | Bandung Wetan |
| 12 | Gempol Park | 1245 | Jl. Gempol | Bandung Wetan |
| 13 | Citarum Park | 1102 | Jl. Citarum | Bandung Wetan |
| 14 | Nyland Park | 783 | Jl. Nyland | Bandung Wetan |
| 15 | Cipunagara Park | 688 | Jl. Cipunagara | Bandung Wetan |

*3.2. Data Collection*

For the first step, we collected all data from Google Maps. The extracted data from Google Maps were used to find the parks' locations. Figure 2 shows the how to find the Google Maps User reviews. Since 2015, Google has seen a more dramatic increase in the number of reviews left compared to other review platforms. Yelp, Facebook, and TripAdvisor have all seen increases in reviews, so this is a positive story for all of them, but Google is growing the fastest by far [53]. This study used an auto-operated web crawler to collect data from Google Maps. We collected reviews based on the names of the parks' locations on Google Maps and then retrieved all the review data from the

users—both ratings and comments on the parks. We collected online reviews from thematic parks and non-thematic parks and used them to compare the extent to which each park is known by visitors and the effectiveness of each park's branding strategy. Online review data collection was conducted in August 2018.

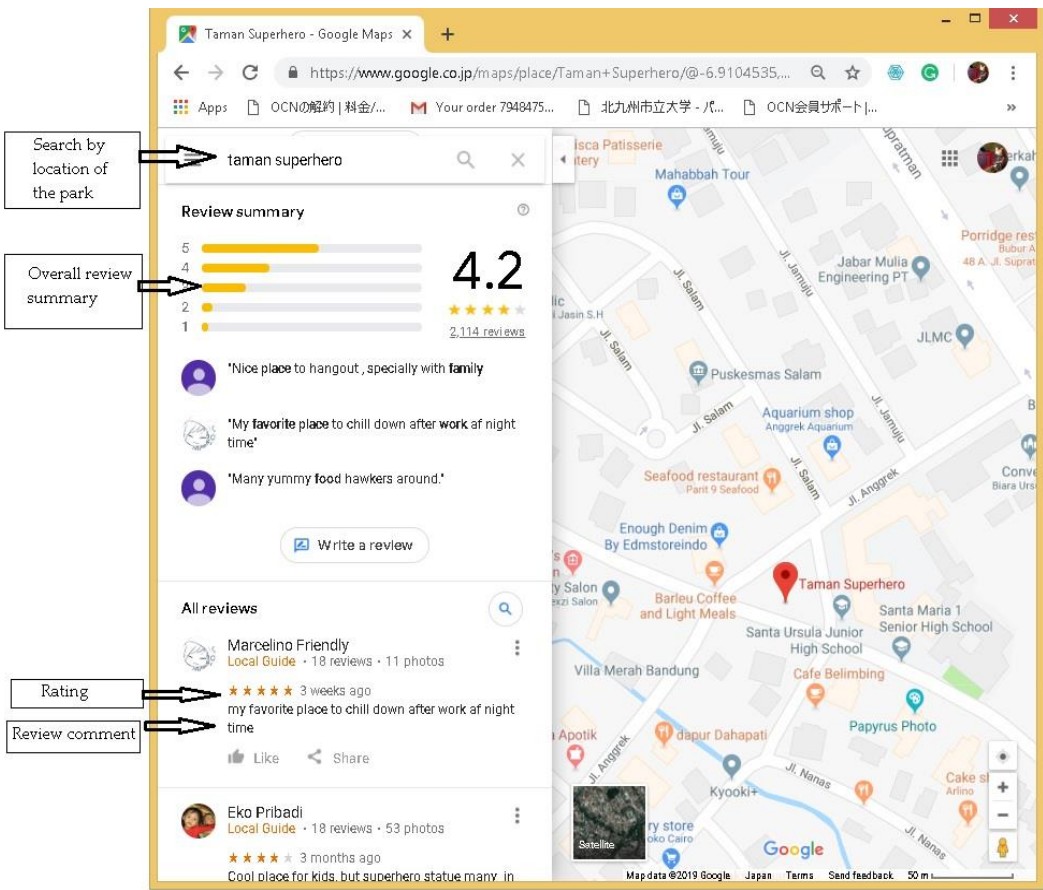

**Figure 2.** An example of Google Maps user reviews.

### 3.3. Text Analysis

Textual analysis is the method of communication that researchers use to describe and interpret the characteristics of recorded or visual messages. The framework of opinion mining of online reviews of thematic parks is shown in Figure 3. The purpose of textual analysis is to describe the content, structure, and functions of the messages contained in text [54]. Text mining, a set of techniques used to discover knowledge and make predictions from text, allows the retrieval of information that is commonly associated with web documents and "text mining techniques are used in web search engines to extract the most relevant documents to the search query". The basic concept behind the retrieval of information is to measure similarity among words, phrases, sentences, and documents [55–57]. Data preprocessing is a critical stage in text mining that is used to transform the initial raw text into a clean dataset. The major steps involved in data preprocessing are word tokenization, stop-word removal, and stemming and lemmatization. Defining what a word means has long been a subject of debate in computational linguistics [58]. A common definition of a word is the smallest unit of meaning [59,60]. By following this definition, the first step of text mining and natural language processing tasks is to segment the input text into linguistic units called tokens. This process is referred to as tokenization [61]. Stop-words are words that provide no information value from a text mining perspective. The main property of stop-words is that they occur extremely frequently. These words are dependent on natural language, and different languages have their own lists of stop-words. Stemming

usually refers to a simple heuristic process that applies a set of rules to an input word in order to remove suffixes and prefixes and to obtain its stem.

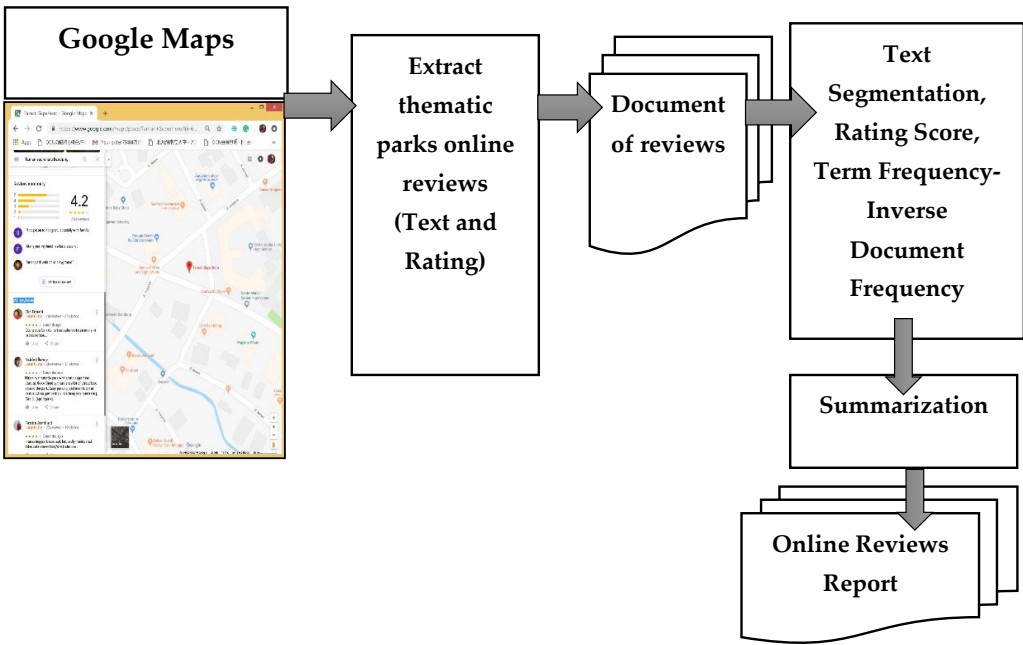

**Figure 3.** Framework of text mining of online reviews of thematic parks.

### 3.4. Term Frequency-Inverse Document Frequency

In this study, we used the Term Frequency-Inverse Documents Frequency (TF-IDF) method to determine the weight of each word. This is a method that is widely used in information search and text mining research. The value of these measurements are used judge the importance of certain terms in certain documents using a collection of several Term Frequency (TF) documents. TF is a value that indicates that certain words are often found in the document—the greater the value, the more important a word in the document is considered to be. Document Frequency (DF) refers to the frequency of use in document collections [62], and the reciprocal number of this value is called the Inverse Document Frequency (IDF), which gives a weight for each word [63]. The IDF is a value that shows how frequently a particular word appears in a document collection. It is the logarithmically scaled inverse fraction of the documents that contain the word, which is obtained by dividing the total number of documents by the number of documents containing the term and then taking the logarithm of that quotient. Therefore, TF-IDF is a value that multiplies TF and IDF and is calculated as follows:

$$TF - IDF = TF \times \log{(N/DF)} \tag{1}$$

where TF is the frequency of a particular word in the documents; N is the total number of documents; DF is the number of documents containing a particular word; and IDF is the inverse of DF.

## 4. Results and Discussion

### 4.1. Comparison of Online Reviews of Thematic Parks and Non-Thematic Parks

We collected data from online reviews from users of Google maps based on the locations of the parks. Figure 4 shows a comparison of the total reviews of thematic parks and non-thematic parks.

The thematic concept provides a special attraction to visitors. The results of the study show that the thematic park concept has a significant impact on the familiarity of visitors with thematic parks compared to non-thematic parks.

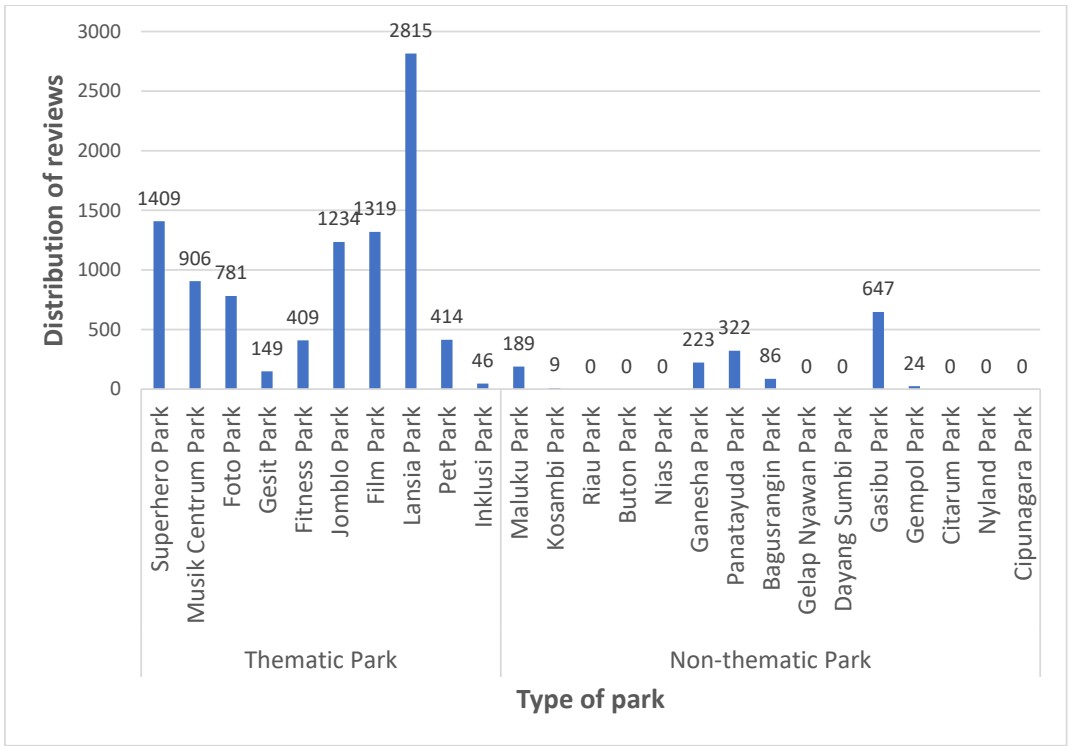

**Figure 4.** Distribution frequency of reviews of thematic parks and non-thematic parks.

*4.2. Review Summary of Thematic Park Rating Distribution*

Score ratings for local places are rated on a scale from 1 to 5 stars. By viewing the locations on Google Maps, we were able to see the Google score, top reviews, and the total number of reviews for each business. Figure 5 shows the review summary based on score ratings.

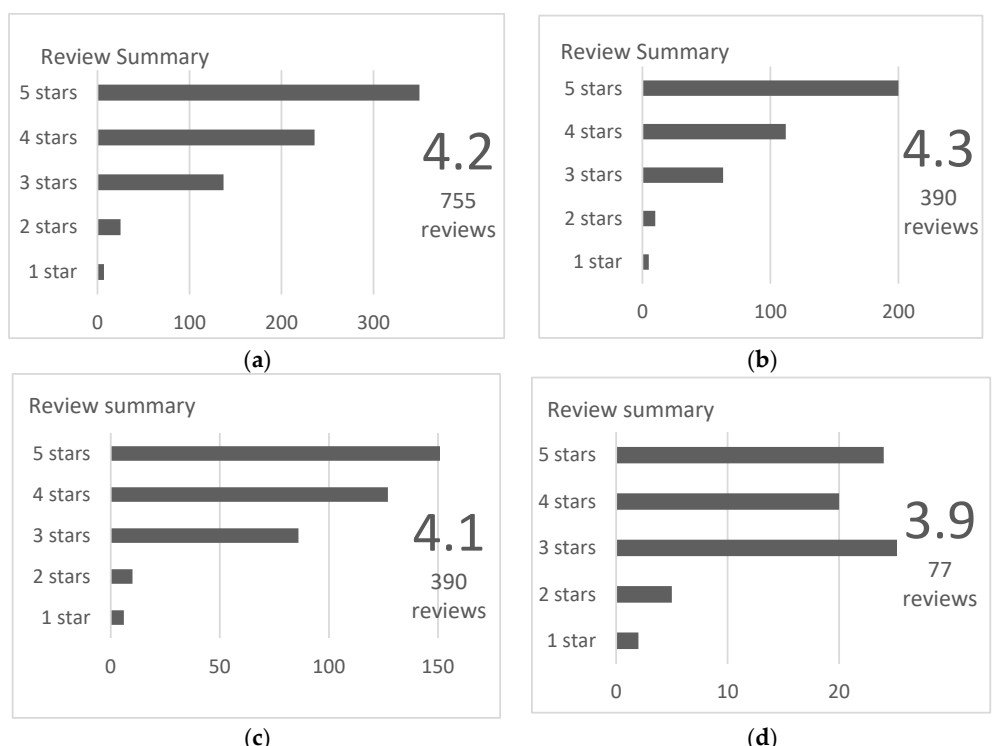

**Figure 5.** *Cont.*

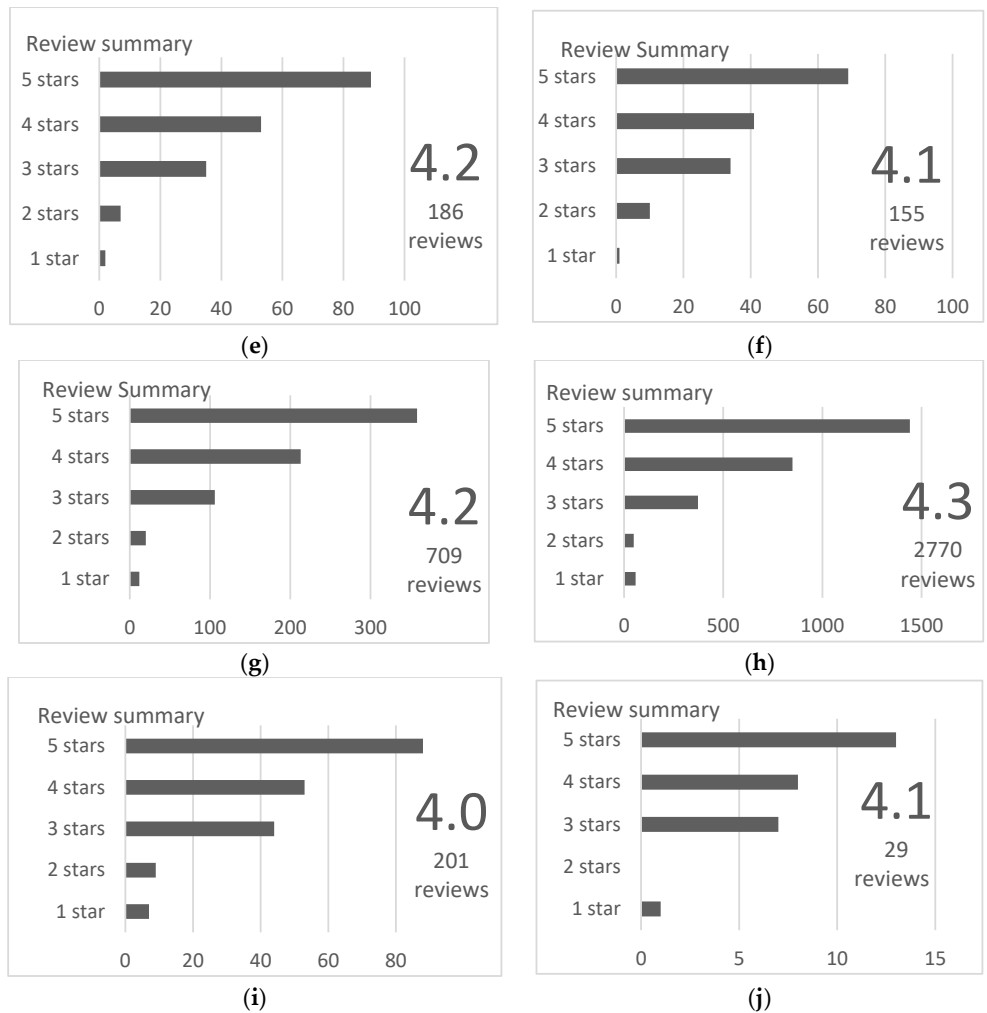

**Figure 5.** Distribution of the online score ratings of ten thematic parks based on the case study: (**a**) Superhero Park, (**b**) Musik Centrum Park, (**c**) Foto Park, (**d**) Gesit Park, (**e**) Fitness Park, (**f**) Jomblo Park, (**g**) Film Park, (**h**) Lansia Park, (**i**) Pet Park, and (**j**) Inklusi Park.

The scores of the parks were represented by user ratings and a variety of other signals. Google's algorithm is designed to extrapolate or estimate the overall rating. The scores are as follows: 5 stars "excellent", 4 stars "very good", 3 stars "average", 2 stars "poor", and 1 star "terrible" [64]. In general, the score is determined from all user reviews, including reviews that only give stars and those with comments. In this research, we only used the review data with stars accompanied by comments. We calculated the score as a weighted average: (total point sum)/(number of voters). The rating distribution of the review summary shows that there are 10 thematic parks with very good ratings (average score: 4.1).

### 4.3. The Term Frequency-Inverse Document Frequency Results

The term frequency-inverse document frequency (TF-IDF) was used to index the significance of each term in a document set. For the online review data collection, we used the comments written by online reviewers. Documents from online review texts were processed to get the frequency of each term in the document. Figure 6 shows the term frequency distribution from 10 thematic parks. The term frequency is a numerical statistic that is intended to reflect how important a word is to a document in the corpus. The identification of the perceptions of reviewers can be calculated from the trends of the terms. Keywords and terms that occur in close proximity are represented as a force-directed network graph, which is shown in Figure 7. The term frequency of each document is shown (see Appendix A).

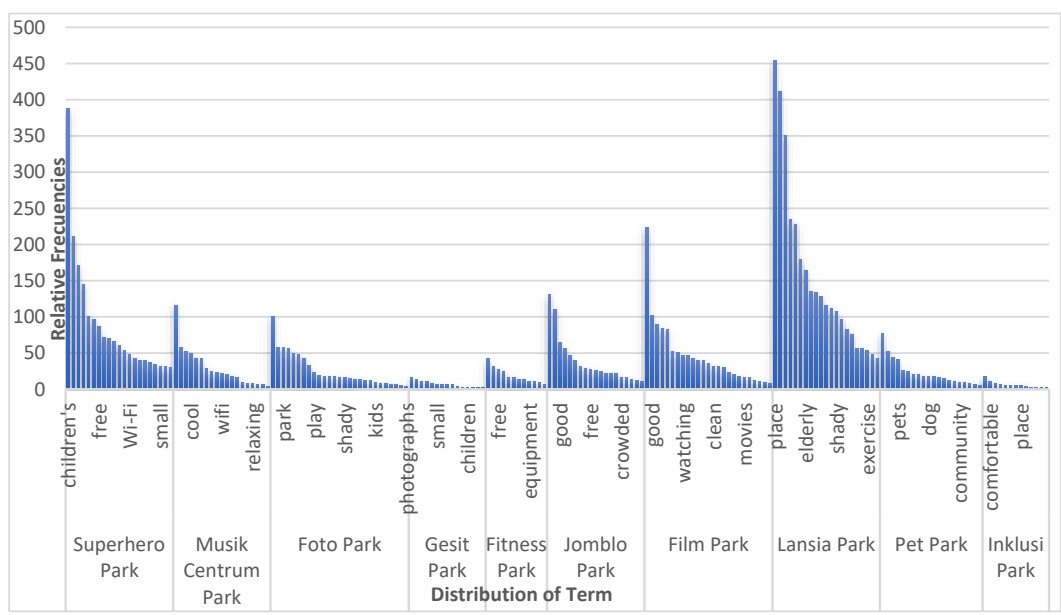

**Figure 6.** Term frequency distribution in online reviews of 10 thematic parks.

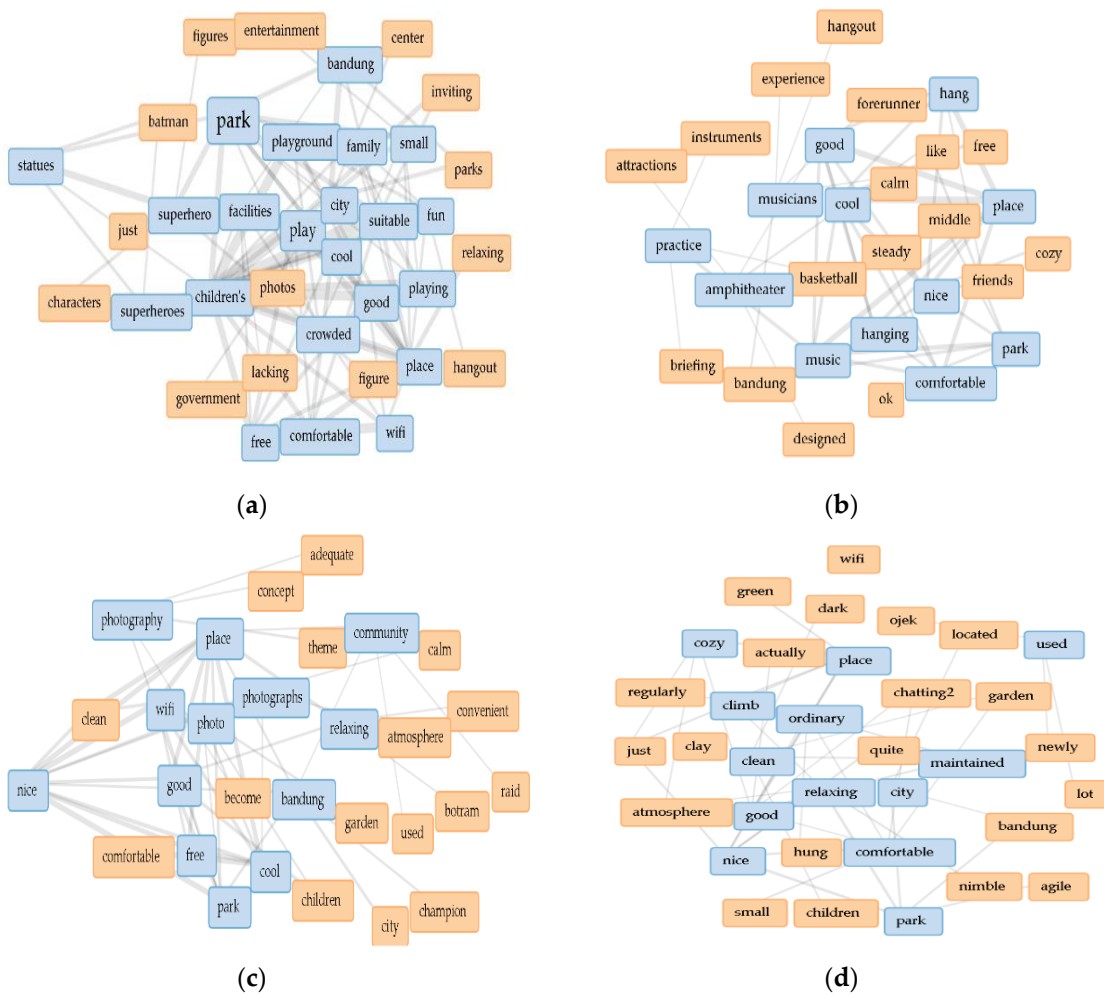

**Figure 7.** *Cont.*

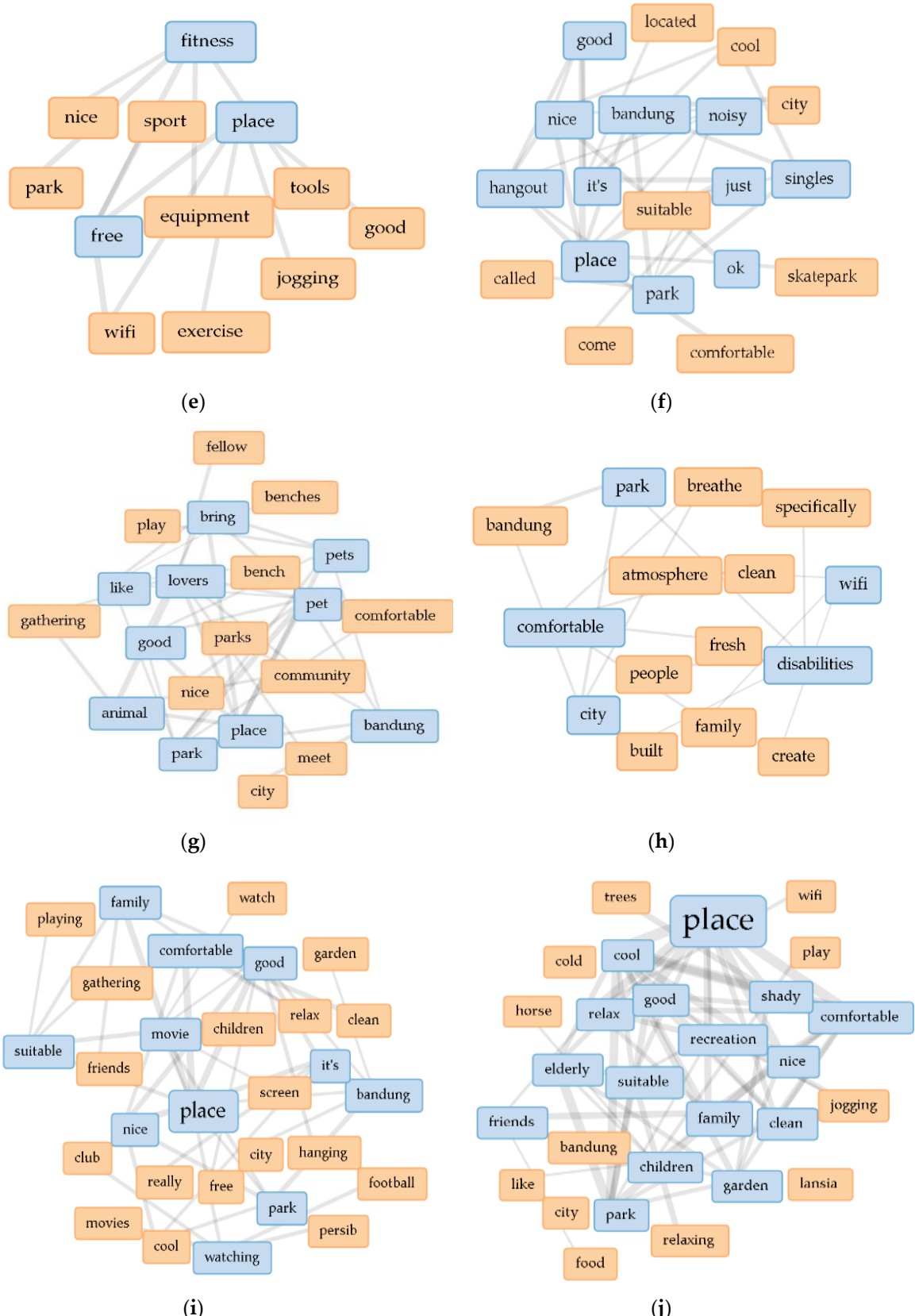

**Figure 7.** Collocation graph of keywords and terms from online reviews.

The network graph shows keywords in blue with links to collocates in orange. This means that each term has a relationship with another term. The text in a particular field of discourse is organized into lexical patterns, which can be visualized as networks of words. Using the relationships among frequency terms, we can summarize the text contained in the online reviews. A summary of the relationships between term keywords in the online review text documents is shown in Table 3.

**Table 3.** Table of summarized keyword relationships from thematic park online reviews.

| Thematic Parks | Summary of Visitors' Perceptions |
| --- | --- |
| Superhero Park | Superhero Park is a place or park that is described as good, suitable, and comfortable for children and families. It has a playground, superhero statues, and free Wi-Fi facilities. |
| Musik Centrum Park | Musik Centrum Park is described as good, cool, nice, comfortable, shady, and suitable for children to play in. It is a musical community where people practice music. |
| Foto Park | Foto Park is a place that is described as nice, good, comfortable, and suitable for families and kids. It has free Wi-Fi and a playground that can be used to play on and take photos. |
| Gesit Park | Gesit Park is a good place that is described as nice, comfortable, small, and suitable for children. It has a playground and free Wi-Fi facilities. It is described as needing maintenance |
| Fitness Park | Fitness Park is a place that is described as nice, with free Wi-Fi and gym equipment, that is shady and is good for sports and jogging. |
| Jomblo Park | Jomblo Park is a good place that is described as nice, unique, and comfortable for young people and single people, with a space for skateboarding, but it is also crowded and noisy. |
| Film Park | Film Park is described as good, cool, nice, comfortable, and suitable for families, children, and the community. It has screen facilities for gathering and watching films or movies, and it has free Wi-Fi facilities. |
| Lansia Park | Lansia Park is a park in Bandung city that is described as good, nice, comfortable, shady, clean, and beautiful. It is suitable for families, children, and for jogging and sports. |
| Pet Park | Pet Park is a good place that is described as nice and comfortable. It is a suitable place for animal lovers to bring pets to and to gather in. It has free Wi-Fi. |
| Inklusi Park | Inklusi Park is a park in Bandung city that is described as cool, clean, and comfortable for families, children, and disabled people. |

## 5. Conclusions

This study confirms that text mining of online reviews can be used to assess visitors' perceptions of thematic parks. We found that the development of thematic parks can be an effective place branding strategy for a city. The large different between the number of reviewers of thematic parks versus non-thematic parks is suggested to reflect the greater attraction of thematic parks. These reviews are likely to increase their popularity, leading to an increased number of visitors. This is definitely in line with the main function of city parks as public spaces and community activity centers. Moreover, the use of visitors' perceptions and reviews for assessing thematic urban parks shows great promise. These assessments provide an overview of the attractiveness of thematic parks and how they are known to the wider community as a type of place branding for the city of Bandung. The conclusion of the online review analysis using text mining is that thematic parks have a greater appeal than non-thematic parks. Thematic parks are better known to the public than non-thematic parks. Ratings from thematic parks were positive with an average rating score of 4.1, which indicates "very good". The term frequency-inverse document frequency (TF-IDF) score-based approach was utilized, and a score was calculated for each review. The summary of the relationships among terms illustrated that thematic parks fulfill the function of providing places for community gathering through particular themes. For example, Superhero Park highlights its function as a children's playground by using superhero statues as a characteristic or attraction of the park. The naming of parks according to their theme and uniqueness is a place branding strategy that influences visitors to share their experiences and perceptions via online reviews. These reviews can then be used as a reference for visitors who are considering visiting thematic parks. A good review and rating will have a major influence on promoting visitors to visit a particular thematic park. Though our research was carefully designed, the conclusions are still subject to some limitations that merit further research. The information provided

in the paper is expected to be used as a reference for the development of thematic parks, especially in the city of Bandung. It can be utilized to design better parks based on the understanding of users' perceptions, and it can be used as a benchmark for similar projects.

**Author Contributions:** This paper was conceived and writing jointly by the authors. M. contributed to the design method and collected and analyzed the data. M.D.K. reviewed the paper, and B.J.D. supervised the research process.

**Funding:** This research received no external funding.

**Acknowledgments:** This study was supported by the University of Kitakyushu, the Ministry of Research, Technology and Higher Education of Indonesia (RISTEKDIKTI), the University of Cokroaminoto Palopo Indonesia.

**Conflicts of Interest:** The authors declare no conflict of interest.

## Appendix A

**Table A1.** Term frequency-inverse document frequency in text reviews of Superhero Park.

| Term | Count | Trend |
| --- | --- | --- |
| children's | 388 | 0.03773952 |
| park | 211 | 0.020523295 |
| place | 171 | 0.016632624 |
| play | 145 | 0.014103686 |
| good | 101 | 0.009823947 |
| suitable | 97 | 0.00943488 |
| free | 86 | 0.008364945 |
| superheroes | 72 | 0.00700321 |
| statues | 70 | 0.006808676 |
| comfortable | 66 | 0.006419609 |
| playground | 60 | 0.005836008 |
| nice | 53 | 0.005155141 |
| Wi-Fi | 48 | 0.004668807 |
| family | 43 | 0.004182472 |
| crowded | 39 | 0.003793405 |
| playing | 39 | 0.003793405 |
| like | 37 | 0.003598872 |
| fun | 34 | 0.003307071 |
| small | 32 | 0.003112538 |
| facilities | 31 | 0.003015271 |
| clean | 30 | 0.002918004 |

**Table A2.** Term frequency-inverse document frequency in text reviews of Musik Centrum Park.

| Term | Count | Trend |
| --- | --- | --- |
| place | 116 | 0.035452 |
| music | 57 | 0.017421 |
| good | 52 | 0.015892 |
| cool | 50 | 0.015281 |
| nice | 43 | 0.013142 |
| hangout | 28 | 0.008557 |
| comfortable | 24 | 0.007335 |
| free | 23 | 0.007029 |
| Wi-Fi | 22 | 0.006724 |
| suitable | 20 | 0.006112 |
| shady | 18 | 0.005501 |
| play | 16 | 0.00489 |
| crowded | 9 | 0.002751 |
| musical | 8 | 0.002445 |
| relaxing | 8 | 0.002445 |
| community | 7 | 0.002139 |
| practice | 6 | 0.001834 |
| musicians | 4 | 0.001222 |

**Table A3.** Term frequency-inverse document frequency in text reviews of Film Park.

| Term | Count | Trend |
|---|---|---|
| place | 101 | 0.029208 |
| nice | 57 | 0.016484 |
| park | 57 | 0.016484 |
| good | 56 | 0.016194 |
| cool | 49 | 0.01417 |
| Wi-Fi | 48 | 0.013881 |
| free | 43 | 0.012435 |
| comfortable | 33 | 0.009543 |
| play | 23 | 0.006651 |
| bandung | 19 | 0.005495 |
| clean | 18 | 0.005205 |
| relax | 18 | 0.005205 |
| photo | 17 | 0.004916 |
| shady | 16 | 0.004627 |
| family | 15 | 0.004338 |
| suitable | 14 | 0.004049 |
| hangout | 13 | 0.003759 |
| facilities | 12 | 0.00347 |
| playground | 12 | 0.00347 |
| kids | 9 | 0.002603 |
| crowded | 8 | 0.002313 |
| relaxing | 8 | 0.002313 |
| photos | 7 | 0.002024 |
| photography | 5 | 0.001446 |

**Table A4.** Term frequency-inverse document frequency in text reviews of Gesit Park.

| Term | Count | Trend |
|---|---|---|
| park | 16 | 0.023155 |
| place | 14 | 0.02026 |
| maintained | 10 | 0.014472 |
| nice | 10 | 0.014472 |
| good | 8 | 0.011577 |
| small | 7 | 0.01013 |
| comfortable | 6 | 0.008683 |
| relaxing | 6 | 0.008683 |
| Wi-Fi | 6 | 0.008683 |
| suitable | 4 | 0.005789 |
| shady | 3 | 0.004342 |
| children | 2 | 0.002894 |
| playground | 2 | 0.002894 |
| facilities | 2 | 0.002894 |
| hangout | 2 | 0.002894 |

**Table A5.** Term frequency-inverse document frequency in text reviews of Fitness Park.

| Term | Count | Trend |
|---|---|---|
| place | 42 | 0.024207 |
| fitness | 32 | 0.018444 |
| free | 27 | 0.015562 |
| good | 24 | 0.013833 |
| nice | 16 | 0.009222 |
| sports | 16 | 0.009222 |
| jogging | 13 | 0.007493 |
| Wi-Fi | 13 | 0.007493 |
| equipment | 11 | 0.00634 |
| gym | 10 | 0.005764 |
| comfortable | 9 | 0.005187 |
| shady | 6 | 0.003458 |

**Table A6.** Term frequency-inverse document frequency in text reviews of Jomblo Park.

| Term | Count | Trend |
| --- | --- | --- |
| place | 131 | 0.023862 |
| park | 110 | 0.020036 |
| good | 64 | 0.011658 |
| singles | 56 | 0.0102 |
| nice | 46 | 0.008379 |
| cool | 39 | 0.007104 |
| comfortable | 32 | 0.005829 |
| Wi-Fi | 29 | 0.005282 |
| free | 27 | 0.004918 |
| hangout | 26 | 0.004736 |
| skate | 24 | 0.004372 |
| single | 22 | 0.004007 |
| suitable | 22 | 0.004007 |
| young | 22 | 0.004007 |
| crowded | 16 | 0.002914 |
| skateboarding | 16 | 0.002914 |
| unique | 13 | 0.002368 |
| flyover | 12 | 0.002186 |
| noisy | 10 | 0.001821 |

**Table A7.** Term frequency-inverse document frequency in text reviews of Film Park.

| Term | Count | Trend |
| --- | --- | --- |
| place | 223 | 0.030286567 |
| good | 102 | 0.013853049 |
| park | 89 | 0.012087464 |
| nice | 84 | 0.011408393 |
| comfortable | 82 | 0.011136765 |
| suitable | 52 | 0.007062339 |
| cool | 51 | 0.006926525 |
| watching | 47 | 0.006383268 |
| family | 46 | 0.006247453 |
| children | 43 | 0.005840011 |
| play | 40 | 0.005432568 |
| film | 39 | 0.005296754 |
| free | 35 | 0.004753497 |
| clean | 32 | 0.004346055 |
| movie | 32 | 0.004346055 |
| screen | 30 | 0.004074426 |
| gathering | 23 | 0.003123727 |
| relaxing | 20 | 0.002716284 |
| kids | 18 | 0.002444656 |
| movies | 16 | 0.002173027 |
| Wi-Fi | 16 | 0.002173027 |
| unique | 12 | 0.001629771 |
| facilities | 10 | 0.001358142 |
| community | 9 | 0.001222328 |
| crowded | 8 | 0.001086514 |

**Table A8.** Term frequency-inverse document frequency in text reviews of Lansia Park.

| Term | Count | Trend |
|------|-------|-------|
| place | 454 | 0.019863 |
| cool | 412 | 0.018025 |
| park | 350 | 0.015313 |
| comfortable | 235 | 0.010281 |
| good | 228 | 0.009975 |
| nice | 179 | 0.007831 |
| elderly | 164 | 0.007175 |
| Bandung | 135 | 0.005906 |
| city | 133 | 0.005819 |
| suitable | 128 | 0.0056 |
| family | 115 | 0.005031 |
| clean | 111 | 0.004856 |
| shady | 108 | 0.004725 |
| jogging | 97 | 0.004244 |
| beautiful | 82 | 0.003588 |
| Wi-Fi | 76 | 0.003325 |
| relaxing | 56 | 0.00245 |
| sports | 56 | 0.00245 |
| exercise | 54 | 0.002363 |
| crowded | 48 | 0.0021 |
| children | 42 | 0.001838 |

**Table A9.** Term frequency-inverse document frequency in text reviews of Pet Park.

| Term | Count | Trend |
|------|-------|-------|
| place | 77 | 0.033304498 |
| pet | 52 | 0.022491349 |
| park | 44 | 0.019031141 |
| pets | 41 | 0.017733565 |
| animal | 26 | 0.011245674 |
| bring | 24 | 0.010380623 |
| good | 21 | 0.009083045 |
| lovers | 21 | 0.009083045 |
| animals | 18 | 0.007785467 |
| dog | 17 | 0.007352941 |
| nice | 17 | 0.007352941 |
| dogs | 16 | 0.006920415 |
| cool | 15 | 0.006487889 |
| play | 12 | 0.005190312 |
| gathering | 10 | 0.00432526 |
| community | 9 | 0.003892734 |
| suitable | 9 | 0.003892734 |
| comfortable | 8 | 0.003460208 |
| facilities | 6 | 0.002595156 |
| Wi-Fi | 5 | 0.00216263 |

**Table A10.** Term frequency-inverse document frequency in text reviews of Inklusi Park.

| Term | Count | Trend |
|------|-------|-------|
| park | 17 | 0.029462738 |
| comfortable | 10 | 0.017331023 |
| city | 8 | 0.013864818 |
| Bandung | 6 | 0.010398613 |
| cool | 5 | 0.008665511 |
| facilities | 5 | 0.008665511 |
| people | 5 | 0.008665511 |
| place | 5 | 0.008665511 |
| disabilities | 4 | 0.006932409 |
| clean | 3 | 0.005199307 |
| relaxing | 3 | 0.005199307 |
| children | 2 | 0.003466205 |
| family | 2 | 0.003466205 |

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
