# Peer review of "Visitor Perceptions and Effectiveness of Place Branding Strategies in Thematic Parks in Bandung City Using Text Mining Based on Google Maps User Reviews"

_sustainability, doi:10.3390/su11072123_

Reviewer 1 Report

The article presents a case study at local scale. Language usage should be improved throughout.

The analysis is relatively poor and underexplained.

Results are described through graphs and maps but should better reported in text.

Discussion should be expanded and conclusions should be reframed in the light of international literature.

The most relevant problem for this article is to clarify the relevance of this study within the international literature on sustainability of urban systems. At the moment, the article is a case study description for relevance only at local scale.

Author Response

Dear Reviewer,

We would like to thank the reviewer for a careful and thorough reading of this manuscript and for the thoughtful comments and constructive suggestions, which help to improve the quality of this manuscript. 

Reviewer 2 Report

Relationship of Thematic Parks in Bandung City Indonesia with Place Branding Strategies Using Text mining Based on Google Maps User Reviews

by Munawir et al.,

The manuscript investigates the importance of thematic parks in city branding by using social media. The results show that thematic parks are much more important and effective in influencing social life than non-thematic ones.

The manuscript is interesting to read. It uses the text mining from Google reviews. There are moderate level problems in English, and presentation quality should be improved. These should be corrected.

There are common problems. Google Map is a proper noun; hence, it should be capitalized. In the captions of figures, text should be lower case and ended with a fullstop. There are some sentences do not have verb!

MINOR COMMENTS:

L17. Oxford comma, please google it (OXC). Insert comma after ‘stories’.

L19-21. Time shift in verb in the same sentence. ……we analyzed….and identify….

L21. Verb missing!

L21. UC (upper case). ….Google Maps user….

L25. LC (lower case). Online reviews….

L40-2. Verb is missing!

L42. I could not get the meaning of image in the parenthesis.

L68. UC. …..Google Maps….

L78. Preposition choice. ….province of Indonesia,…..

L79. Superscript. ……km2.

L81. Comma after plan. After a noun phrase generally longer than four words, please include comma. Look at L92.

L89. LC. Thematic park….

L91-3. Re-write this sentence. Simply write: Based on XXXX, there are  631 parks in the city [REFS].

L93. Comma before ‘and’.

L96. Detail data …….. are shown in Table 1 and Table 2. Figure 1 shows……

L97. Caption. LC. …..thematic parks…

L97. In the description of Superhero Park, you mentioned one statue of superhero; however, there are several in this park mentioned in L 260. Please make sure consistency!

L97. Description of Musik Park. Insert comma after ‘art’.

L97. Description o Foto Park. Complete the sentence with a ‘fullstop’.

L97. Description of Pet Park. UC. Inklusi Park is one…..

                                    Also delete ‘a city’. Repetition!

L99. Caption. LC. ….non-thematic parks…

L150. Comma before ‘and’.

L153. UC. …in Figure 3.

L154. Complete the caption with a ‘fullstop’. Also for Figure 4.

L174. Capitalization.

L178. Use lower case.

L209. Use lower case.

L209. Improve Figure 5. Make sure it within one page!

L217. …Figure 4.

L218. Comma after ‘term’.

L220. LC. …of term frequency…

L247. LC. …of summarization …..thematic park….

L247. Summary of Superhero Park. Add ‘fullstop’ to many of them! Foto Park, Fitness Park, Film Park, Lansia Park, and Pet Park.

Pay attention summary of Pet Park. Re-write it!

L256. Comma after ‘utilized’.

L274. Comma after ‘paper’.

L276. Include ‘fullstop’.

Author Response

(The authors gave the same response as above.)

Reviewer 3 Report

Dear Authors,

Thank you for submitting this paper. I enjoyed reading it. 

I have a few revisions, one of them is a bit more substantial. First, you do not have a 'background' or 'literature review section'...why? I recommend to include this. This is essential to get an updated understanding on the topic, in this case about parks, thematic parks and the contrast with traditional non-thematic parks. In this discussion you can also include something about 'public space' for instance as some thematic parks are already privatised! I think this section should be done, otherwise the paper and its contents lacks of theoretical framework. 

Secondly, understanding tables and figures can be improved if you locate them 'after' the text that announces them, not before. A text that introduces them is necessary. This text should indicate what we need to see in the table but not in descriptive terms (that information is already in the table, so, not necessary to repeat or paraphrase what we see by ourselves). So, please add introductory-analytical short texts before tables and figures to understand better their purposes. 

I think this is a good paper, very clear and, methodologically substantial but incomplete as it does not have a theoretical framework! so, please add it (maybe after the Introduction, but check the journal norms first to be sure) and it will be a more completed proper paper. In this present condition, I cannot support its publication.

Thank you,

Author Response

(The authors gave the same response as above.)

Reviewer 4 Report

The paper presents an interesting topic which fits to the scope of journal. However, the current version of the paper contains crucial disadvantages, and in my opinion should not be published.

The text does not present a coherent story. In some parts of the text there is no clear cause-effect flow between sentences. In current version reader has to structure some information on his own, which is unacceptable.

There are basic grammar mistakes in many sentences, which does not allow to interpret the meaning, or allow to understand in few different ways.

Lines 27-29: the results can be used for „sustainable city parks”. What does it mean? It is useful for parks which are already sustainable? Or to make parks sustainable in the future? The Authors seem to be not very careful with the concepts and expressions that they are using.

There is no clearly defined gap of knowledge, and based on that, the aim of the study. Lack of clearly defined aim of the research does not allow the reader to answer in the conclusions, if the aim was achieved or not. Current goal in lines 74-75 does not constitute such aim.

Graph in figure 3 is linear, however, in current version it looks like a schematic block diagram with more complicated structure. All arrows and location of blocks should intuitively inform the reader about the correct way to read it. Moreover, the quality of presentation is low.

The linear chart suggest the reader connections with following objects and their values. Figure 4 and figure 6 should use rather bar charts. Moreover, the chart in figure 6 is not readable.

The figure with frequency of the used terms does not include all terms (probably because of the size of the figure). Nevertheless, the chosen way of data representation does not allow to verify the result. However, in one example (Superhero Park) I noticed that term “children’s” was used more often that “play”. But on the graph the term “play” is written in bigger fonts. Such way of data representation makes the reader concerned about the proper interpretation.

Line 217: should be “shown in figure 7”.

The text nor the legend describe the difference between blue and orange blocks in graphs (figure 7). Probably the colors refer to the centrality of the objects, however, it is not explained.

Conclusions start with the sentence “xxx”, however, due to the aim of the research (which I commented previously) the verification of the method was not a goal of the study. Therefore, those elements are not consistent.

Line 247: it is the third table in this text.

The numbering of references are not correct according to the MDPI style. In line 47 the reference number should be 6 instead of 37. In next line there is reference number 18 instead of 7, etc.

Words beginning with block capitals within some sentences should be changed to lower capitals, e.g. line 25 “Reviews”, line 56 “Large”, line 63 “Identification”, etc.

Author Response

Dear Reviewer,

We would like to thank the reviewer for a careful and thorough reading of this manuscript and for the thoughtful comments and constructive suggestions, which help to improve the quality of this manuscript.

Round  2

Reviewer 1 Report

good revision overall. Many points revised. The article is now clean and standardized. Language usage should be improved throughout.

Author Response

Dear Reviewer 1,

Thank you very much, we appreciate the positive feedback from the reviewer.

The authors would like to thank the area editor and the reviewers for their precious time and invaluable comments. We have carefully addressed all the comments.

Reviewer 2 Report

Thanks to the authors for their effort to repsond my comments.

Author Response

Dear Reviewer 2,

Thank you very much, we appreciate the positive feedback from the reviewer.

The authors would like to thank the area editor and the reviewers for their precious time and invaluable comments. We have carefully addressed all the comments.

Reviewer 3 Report

Dear authors,

Thank you for undertaking the suggested revisions.

However, I have to insist that you have not addressed the suggested inclusion of providing a literature review section. This is relevant as it contextualises the paper in the international debate of the research topic. I insist that a 'literature review' section should be done. At the moment, you argued that in the introduction section you have placed this but is not. There are several terms that are used (such as thematic parks, place branding, place attachment, people's perceptions, etc.) that makes the introduction confusing in terms of focus and these subjects are not in-depth discussed and used to build the paper's argument. In my view, I insist that all of these ideas should discussed in a separated 'literature review section' for the understanding of 'thematic parks'. Here, you can critically review current definitions, planning approaches, understandings and how they resonate with the international context and literature. This is relevant to understand 'why' this paper needs to be published? What is the contribution to the extant knowledge and in what? (thematic parks? Place attachment? , etc.). What is the research gap and questions? etc. In that sense, I urge you to include this section as the paper has valuable empirical material but does not have a theoretical framework. On this basis, I cannot support the publication for this moment. 

There are still some inconsistencies in introducing figures and tables 'before' showing them. 

Although I am not a native English speaker, I can still see some language inconsistencies. Please revise that for a new version.

Author Response

Dear Reviewer 3,

Thank you very much, we appreciate the positive feedback from the reviewer.

The authors would like to thank the area editor and the reviewers for their precious time and invaluable comments. We have carefully addressed all the comments.

Reviewer 4 Report

I still doubt about the quality of the paper.

The aim defined by the Authors is as follows: “The aims of this study are to identify the effectiveness of thematic parks in creating a branding of the places in the city of Bandung and to know the perceptions of the community about thematic parks through social networks.”

According to Oxford dictionaries “effectiveness is the degree to which something is successful in producing a desired result; success; e.g. ‘the effectiveness of the treatment’ “.

I understand that thematic parks in Bandung might have also the aspect of city branding. However, was it the main desired result of the project of constructing these parks? Moreover, quite often “effectiveness” in research include resources that were used in relation to results that were achieved. If in two cases we achieve the same result, but in case A we have to pay $1, and in case B we have to pay 2$, than actions in case A are two times more effective.

If the Authors want to follow that concept, I would suggest to include the aspect of resources that were spend in order to construct thematic parks in Bandung. Moreover, as the branding does not seem to me as the main aspect of urban greenery, I would highlight it somewhere (in introduction or in the discussion) which factors of sustainable urbanization were improved – next to social aspects which were mentioned in the introduction, I miss environmental dimension of that investment, see for instance: “Spatial Form of Greenery in Strategic Environmental Management in the Context of Urban Adaptation to Climate Change” which discuss the impact of the urban greenery in context of environmental impact on cities.

Author Response

Dear Reviewer 4,

Thank you very much, we appreciate the positive feedback from the reviewer.

The authors would like to thank the area editor and the reviewers for their precious time and invaluable comments. We have carefully addressed all the comments.

Round  3

Reviewer 3 Report

Dear Authors,

Thank you for undertaking revisions to the manuscript, including a literature review section. 

However, the literature review section is still narrow. Although you placed the relevant themes (thematic parks, place branding and visitors perception) these sections are too descriptive, short, and barely discuss the argument behind these themes. If literature review is only about definitions, it could be enough to cite previous works and this is not the case. Therefore, literature review must be further expanded and provide a critical revision of the extant knowledge in the field and what is still less addressed. This is something that is still lacking.

I can see the draft have valuable empirical material, but without proper contextualisation in the extant knowledge, it is difficult to see what is the paper's contribution. I suggest to do the literature review again in consideration of this. So, provide a more analytical and critical literature review section. I was searching on Google scholar and there is plenty of updated information about thematic parks and its related issues. 

In this sense and although I recognise the empirical values of the paper, I cannot approve it before making these revisions. 

One more suggestion: response letters should be more explicative of the revisions. It is not enough to send a message like 'revisions are done' to then suppose that because you have written something that would be enough. Please, see how response letters are done because they discuss the suggestions in light of the new information and argumentation added. That information is possible to be found in the Internet.

Author Response

Dear Reviewer, 

We appreciate the positive feedback from the reviewer.

The authors would like to thank the area editor and the reviewers for their precious time and invaluable comments. We have carefully addressed all the comments.

Reviewer 4 Report

My main concern in the second review was the aim of the research, namely, using the word „effectiveness”. As I already mentioned, effectiveness refers to resources that were used in relation to results that were achieved. However, I do not see any progress in that aspect.

According to changes in literature review, the first thing that revised paper is not prepared in track changes, which force me to compare it with previous versions “manually”. Putting the comments show only “where” but not “which” changes appear in the text.

New sentences are often not grammatically correct, like: “The benefits and functions of the ecosystem green space for humans could be directly or indirectly.” Should be “directly” or “direct”?

Author Response

(The authors gave the same response as above.)
